# Direct Cost Analysis of Microbial Keratitis in North China: A Hospital-Based Retrospective Study

**DOI:** 10.3390/pathogens13080666

**Published:** 2024-08-07

**Authors:** Qingquan Shi, Bo Peng, Zhen Cheng, Zijun Zhang, Zhenyu Wei, Zhiqun Wang, Yang Zhang, Kexin Chen, Xizhan Xu, Xinxin Lu, Kai Cao, Xueyao Wei, Qingfeng Liang

**Affiliations:** Beijing Institute of Ophthalmology, Beijing Tongren Eye Center, Beijing Tongren Hospital, Capital Medical University, Beijing 100005, China; qingquan@mail.ccmu.edu.cn (Q.S.); 13980597816@163.com (B.P.); cz193172709cz@163.com (Z.C.); shenyu@ccmu.edu.cn (Z.Z.); weizhenyu@ccmu.edu.cn (Z.W.); eyewzq@163.com (Z.W.); biozy1@126.com (Y.Z.); ckxwl1234@hotmail.com (K.C.); xuxz0924@mail.ccmu.edu.cn (X.X.); luxinxin2009@126.com (X.L.); anzhen602@163.com (K.C.); 13720000174@163.com (X.W.)

**Keywords:** microbial keratitis, corneal ulcer, healthcare cost, economic burden, China

## Abstract

Microbial keratitis (MK) is the fourth leading cause of blindness globally, imposing a substantial burden on the healthcare system. This study aims to determine the cost composition of MK patients and explore factors influencing these expenses. We analyzed the demographics, clinical features, and costs of 602 MK patients treated at Beijing Tongren Hospital from June 2021 to October 2023. The analysis revealed the average total cost of treating MK was USD 1646.8, with a median of USD 550.3 (IQR: 333.3–1239.1). Patients with Acanthamoeba keratitis (AK) incurred the highest median total costs at USD 706.2 (IQR: 399.2–3370.2). Additionally, AK patients faced the highest costs for ophthalmic exams and laboratory tests (both *p* < 0.001), while patients with fungal keratitis (FK) and viral keratitis (VK) experienced higher medication costs. Costs varied significantly with the severity of MK, especially for outpatients at severity level 4, which was markedly higher than levels 1–3 (USD 1520.1 vs. USD 401.0, *p* < 0.001). Delayed presentation also resulted in increased costs (USD 385.2 vs. USD 600.3, *p* < 0.001). Our study highlights the financial burden associated with MK treatment and underscores the importance of timely and accurate diagnosis and intervention.

## 1. Introduction

Microbial keratitis (MK), often described as a “silent pandemic”, poses a significant public health challenge globally [1]. Ranked as the fourth leading cause of blindness globally, MK imposes a substantial burden on healthcare, with a notable estimation of from 1.5 to 2 million cases resulting in monocular blindness. Strikingly, up to 90% of these cases occur within developing nations [2,3,4]. In the US, it resulted in approximately 1 million medical visits in 2010, with an annual cost of around USD 175 million [1]. In the UK, emergency hospital admissions due to MK incur an average cost of GBP 2855 [5]. In developing countries, there are not many records of the expenses for MK, but a study by a tertiary referral hospital in Thailand showed that the median cost for hospitalized MK patients was USD 10,840, highlighting a noteworthy economic impact on society [6].

MK is a complicated disease with four main types: bacterial keratitis (BK), fungal keratitis (FK), viral keratitis (VK), and Acanthamoeba keratitis (AK) [7]. Each type requires unique diagnosis and treatment methods [2,8]. Despite existing guidelines, management practices vary among specialists [9]. Factors such as patient age, gender, systemic diseases, time to presentation, and adherence to treatment significantly influence disease progression and outcomes [10]. These variables can affect the total cost of managing MK, with costs fluctuating based on the type of keratitis and other parameters. However, there is a lack of in-depth research on these cost-related factors, particularly in the context of China

Thus, understanding MK costs is crucial for patient care and public health strategies and the analysis of specific cost-related factors enables tailored treatment leading to reduced financial strain on patients and society. This study seeks to estimate the cost associated with MK in China, shedding light on its economic ramifications for both individuals and the healthcare system. It includes an analysis of medical records and cost data from patients at Beijing Tongren Hospital, which is the premier ophthalmic center in North China. Additionally, the research examines the influence of various factors on the patients’ costs.

## 2. Materials and Methods

### 2.1. Study Design

This study was a retrospective, single-center hospital-based analysis, and the research procedure was approved by the Medical Ethics Committee of Beijing Tongren Hospital (TRECKY2021-024). The study protocol fully adhered to the Declaration of Helsinki and the ARVO statement on human subjects. Patients diagnosed with MK who presented to Beijing Tongren Hospital between June 2021 and October 2023 were included. The data including age, gender, time to presentation, number of visits, number of hospitalizations, risk factors related to MK, clinical symptoms and signs, and costs for MK treatment, were collected and analyzed.

### 2.2. Patients

Patients enrolled had to meet at least one of the following eligibility criteria: (1) BK, FK, or AK were defined as having compatible clinical manifestations and at least one positive laboratory test (smear or microbial cultures). (2) VK had to be diagnosed by three cornea specialists based on their medical history, clinical manifestations, and the efficacy of antiviral treatment, or in accordance with positive PCR test results [2]. Patients were excluded if they had one of the following criteria: (1) patients with other concurrent eye diseases currently (such as neurotrophic keratitis, autoimmune eye diseases, nasolacrimal duct inflammation, etc.); (2) patients lacking slit-lamp images or detailed clinical description; (3) patients with incomplete follow-up data affecting prognostic evaluation.

For BK, FK, and AK patients, they must complete at least two follow-up visits, and the final follow-up is determined under three conditions: (1) corneal infiltration has turned into a scar; (2) a significant improvement in the keratitis condition based on the patient’s external eye image assessment at the time of the last treatment; (3) the patient has undergone a corneal transplant without significant immune rejection or recurrence of infectious keratitis. For VK patients, the final follow-up is defined as either sustained resolution of corneal infiltration into a corneal scar or the absence of significant immune rejection after a corneal transplant, with no recurrence for at least 12 months and an improvement in visual acuity [11].

### 2.3. Clinical Evaluation and Microbiological Examinations

This research compiled data on risk factors and clinical manifestations of MK. Risk factors included a history of contact lens use, corneal trauma, or previous eye conditions like uveitis, glaucoma, and cataracts. Diabetes and hypertension were also noted as potential risk factors. Clinical symptoms reported by patients included redness of the eye, eye pain, blurred vision, tearing, and sensitivity to light. Clinical signs documented in medical records included corneal infiltration, ulcers, edema, neovascularization, lesions involving the central cornea, and hypopyon in the anterior chamber.

Patients were categorized into four severity levels using two recognized scales: Keay et al. [12] and Acharya et al. [13]. Level 1 indicated mild cases with lesions outside the central 4 mm area. Level 2 signified moderate cases with lesions partly involving the central 4 mm zone with a diameter smaller than 2 mm outside. Level 3 denoted severe cases with lesions involving the central 4 mm zone with a diameter larger than 2 mm. Level 4 indicated the most severe cases with lesions covering the central 4 mm zone.

Specimens from the cornea were collected by scraping the base and edges of the ulcer with a platinum spatula under a slit-lamp microscope for microbiological testing. For suspected BK, FK, and AK patients, the samples were placed on glass slides for staining with Gram and Giemsa stains. Blood agar medium, chocolate agar medium, potato dextrose medium, and Page’s medium with Escherichia coli were used to culture microorganisms from corneal lesions, including bacteria, fungi, and Acanthamoeba. Mass spectrometry and sequencing techniques were utilized to identify some rare, isolated strains grown from the medium. In cases of suspected viral keratitis, a sample was collected from the corneal lesion and examined through a comprehensive RT-PCR test. This assay was designed to simultaneously detect the presence of viral DNA from key pathogens including Adenovirus, Herpes Simplex Virus types 1 and 2 (HSV-1, HSV-2), Varicella–Zoster Virus (VZV), and Cytomegalovirus (CMV), ensuring rapid and precise diagnosis that facilitates immediate initiation of appropriate therapy to safeguard the patient’s vision [2,14]. Before obtaining definitive microbiological test results, specialists will initiate empirical therapy with antibiotics, antivirals, or anti-amoebic medications for the patients. Upon receipt of the microbiological test results or PCR findings for viruses, the treatment plan will be adjusted by the specialists to provide targeted, specialized treatment accordingly.

### 2.4. Direct Costs Data

This study analyzed both total and average costs per visit for each MK patient treated at Beijing Tongren Hospital. For one patient, the total cost was calculated as the sum of all expenses during outpatient and inpatient visits. The average cost was calculated by dividing the total costs by the total number of visits, which is the sum of outpatient and inpatient visits. The median total costs for each condition were essentially the median of the total costs for all patients. Detailed expense data were obtained from the hospital’s information department, including costs for registration, ophthalmic exams, lab tests, medication, cornea surgery, minor procedures, medical consumables, and other costs. The definition of each type of service payment is outlined in Table 1. For each MK patient, we aggregated the various types of expenses and calculated the costs of each type to be used in subsequent analyses. All costs were originally calculated in RMB (¥) and converted to US dollars ($) based on the 2023 exchange rate of 1 yuan = 0.14 US dollars. 

### 2.5. Statistical Analysis

In this study, statistical analysis was conducted using R version 4.3.2 (The R Foundation Vienna, Austria). Demographic data of patients were examined using chi-square tests. The total and average costs of MK outpatients were analyzed with Mann–Whitney U tests. Comparisons between single MK and mixed MK patients were made using Mann–Whitney U tests. Total costs and different service payments were analyzed under different influencing factors, including age statistics using the Kruskal–Wallis’s test and Dunn’s test, and other analyses with the Mann–Whitney U test. The relationship between outpatient and inpatient visits and total costs was assessed using multiple linear regression analysis. The correlation between demographics and costs was examined using the Spearman correlation coefficient (for continuous variables vs. continuous variables), point-biserial correlation coefficient (for dichotomous variables vs. continuous variables), and Cramér’s V (for categorical variables vs. categorical variables). Significance was defined as *p* < 0.05, denoted by (*) in the tables. The symbols marked on the image indicate statistical significance levels: * represents 0.01 < *p* < 0.05, ** represents 0.001 < *p* < 0.01, *** represents *p* < 0.001.

## 3. Results

### 3.1. Demographics and Clinical Features

This study included a total of 602 patients diagnosed with MK. The demographic and clinical characteristics of these patients are summarized in Table 2. Of the cases, 375 (62.3%) males and 227 (37.7%) females were included, with a median age of 57 (IQR: 46–65). In addition to the median time to presentation at this hospital of 20 days (IQR: 4–60 days), patients had a median number of total visits of five times (IQR: 3–9 times). Among the patients with MK, 294 (48.8%) had infections in the left eye, 339 (56.3%) in the right eye, and 31 patients (5.1%) had infections in both eyes. Additionally, for 251 patients with positive culture results (positive PCR results for viruses), the results for bacteria, fungi, and viruses are shown in Appendix A. Gram-positive bacteria were more frequently detected (48 cases) compared to Gram-negative bacteria (31 cases), with Staphylococcus epidermidis being the most common. Fungal culture included 113 positive results, with molds significantly more common than yeasts. Among viruses, Herpes Simplex Virus Type I was the most common (five cases), followed by Epstein–Barr virus (four cases). Acanthamoeba species were the only parasites identified, totaling 48 cases.

Among patients with single microbial keratitis (SMK), BK patients were the most prevalent, comprising 251 cases (41.7%), followed by FK with 128 cases (21.3%). There were also 42 (7.0%) cases of VK and 48 (8.0%) cases of AK. Additionally, mixed microbial keratitis (MMK) was observed in 133 cases (22.1%), and the specific types are detailed in Appendix A. Of the 602 patients, 251 had positive culture or PCR results, summarized in Appendix A. Gram-positive bacteria were more frequent (48 cases) than Gram-negative bacteria (31 cases), with *Staphylococcus epidermidis* being the most common. Fungal cultures yielded 113 positive results, with molds (especially *Fusarium* spp.) significantly more common than yeasts. Among viruses, Herpes Simplex Virus Type I was the most frequent (five cases), followed by Epstein–Barr virus (four cases). *Acanthamoeba* spp. were the only parasites identified, totaling 48 cases.

The age difference among SMK patients was statistically significant (*p* < 0.001). Notably, AK patients were significantly younger than those with other types of keratitis. The median time to presentation for patients with BK, FK, VK, and AK was 13, 18, 50, and 30 days, respectively, showing a significant difference (*p* < 0.001). AK patients had the highest median number of visits, followed by those with VK, BK, and FK, indicating a significant statistical difference (*p* = 0.003). For hospitalization rates, FK patients were highest (28.1%), followed by AK (20.8%), BK (12.7%), and VK (11.9%) patients (*p* = 0.002). Compared with SMK, patients diagnosed with MMK showed significantly higher median age and number of visits (*p* < 0.001 and *p* = 0.032, respectively). Furthermore, hospitalization rates were elevated among MMK patients (*p* = 0.037). 

### 3.2. Direct Cost Analysis

An analysis of direct costs involving 602 MK patients showed an average total cost of USD 1646.8, with a median of USD 550.3 (IQR: 333.3–1239.1). For each type of keratitis, the median total costs were as follows: BK at USD 419.0 (IQR: 281.7–815.8), FK at USD 610.8 (IQR: 376.1–1814.1), VK at USD 577.1 (IQR: 376.1–970.1), and AK at USD 706.2 (IQR: 399.2–3370.2), which demonstrated significant differences (*p* < 0.001), with AK displaying the highest median total cost, indicating statistical significance. Among outpatient costs for diagnosis and treatment, the median total costs were USD 539.2 (IQR: 332.9–967.7). The median total outpatient costs of BK, FK, VK, and AK patients were as follows: USD 419.0 (IQR: 281.4–750.2), USD 571.5 (IQR: 376.5–1031.9), USD 577.1 (IQR: 377.1–927.4), and USD 703.7 (IQR: 399.2–1380.3), indicating significant difference (*p* < 0.001) (Table 3). Specifically, considering the severity levels of the MK, those with a severity level of 4 incurred significantly higher costs than patients with severity levels 1–3 (USD 1520.1 (IQR: 662.6–6054.2) vs. USD 401.0 (IQR: 280.8–605.5), *p* < 0.001). Among inpatient costs for diagnosis and treatment, the median total costs were USD 4683.9 (IQR: 1173.1–5029.6). The median total costs of BK, FK, VK, AK patients were as follows: USD 4403.4 (IQR: 1230.6–4822.2), USD 4870.1 (IQR: 513.2–5855.8), USD 4753.5 (IQR: 4696.7–4939.4) and USD 4716.6 (IQR: 4405.5–4985.6), indicating no significant difference (*p* = 0.398). 

Both outpatient and inpatient costs for different types of keratitis were analyzed (Figure 1 and Appendix A), and median costs of various keratitis types were displayed. In inpatient care, significant differences between keratitis types were found only in medication and lab test costs among keratitis types (*p* < 0.001 and *p* = 0.032, respectively). In outpatient care, core costs included registration, ophthalmic exams, lab tests, and medication costs. Regarding ophthalmic exams and lab test costs, the median for AK patients was significantly higher than those for the other three types of patients, indicating statistical significance (both *p* < 0.001). As for medication costs, patients with VK had the highest median, followed by FK and AK. BK had the lowest median with statistically significant differences (*p* < 0.001). For registration costs, patients with SMK showed significantly lower costs compared to those with MMK (*p* = 0.005). Additionally, the median ophthalmic exams and lab test costs for patients with mixed microbial keratitis were significantly higher than those for SMK patients (both *p* = 0.003).

### 3.3. Analysis of Key Factors Influencing Costs 

Some factors that affected the total costs for MK patients are listed in Table 4, including gender, age, time to presentation, and the presence of systemic diseases (hypertension, diabetes). Comparisons of the total costs under each influencing factor were made and presented. The statistical results indicated that there were no significant differences in total costs across different genders, age groups, or the presence of hypertension and diabetes. However, it demonstrated a wide discrepancy in total costs between patients who presented within or after one week (USD 385.2 (IQR: 298.3–686.1) vs. USD 600.3 (IQR:352.7–1660.7), *p* < 0.001), indicating statistical significance.

In Figure 2, patients with a presentation time longer than a week have significantly higher costs in registration, ophthalmic exams and lab tests, and medication costs compared to patients presented within a week, also indicating statistical significance (*p* = 0.031, *p* = 0.021, *p* = 0.037, *p* < 0.001). 

Correlations between demographic information and various types of costs are presented in Figure 3. The results indicate significant correlations, where age significantly correlates with hospitalization status, time to presentation, and severity level; the number of visits shows significant correlations with hospitalization status, severity level, total costs, and all types of costs; time to presentation is significantly correlated with total and average costs, as well as registration and medication costs; severity level has significant correlations with all types of costs; and all costs data (total costs, average costs, and each type of costs) are significantly correlated with each other. 

Multivariate linear regression analysis was utilized to examine the impact of multiple parameters on total costs. The results demonstrate a strong linear relationship between several parameters and total costs within the fitted model (adjusted R2 = 0.672, *p* < 0.001). Among these, the type of disease as AK, the number of outpatient visits and inpatient visits, and most severe cases (severity level = 4) were significantly associated with an increase in total costs (*p* = 0.028, *p* < 0.001, *p* < 0.001, *p* = 0.003 respectively). Among them, the number of outpatient and inpatient visits are the two most important influencing parameters (standardized coefficient: 0.245 and 0.595). Analyzing the regression coefficients for the number of outpatient visits and the number of inpatient visits revealed that each additional outpatient visits result in an increase of USD 100.3 in total costs, while each additional inpatient visit leads to an increase of USD 2759.6 in total costs on average (Table 5).

## 4. Discussion

This is the first study to analyze the costs of MK in China, investigating expenses across different MK types and the factors influencing these costs. The findings show that MK creates a heavy financial burden on patients, especially for AK patients who have the highest median total costs among MK types. Additionally, AK patients bear the highest costs of ophthalmic exams and lab tests. The severity of MK is directly linked to higher total costs. The timing of hospital visits significantly increases these costs and outpatient expenses are considerably lower than inpatient costs.

According to our study, the average total cost per patient is USD 1646.8. Compared with China’s average disposable income of USD 5490.5 in 2023, the typical MK patient would need to allocate about 3.6 months’ worth of income if opting for a fully self-funded payment method. According to a survey from Thailand in 2023, the average cost for MK patients was USD 140.1 [15]; in the US, the average cost for MK was USD 1788.7 [16]. These differences may relate to local health insurance, doctor expertise, and patient education, etc. However, the comparison of healthcare costs among countries is challenging [10].

The cost analysis for patients with four types of keratitis shows that AK patients had the highest median total cost, followed by FK and VK, with BK patients having the lowest median total cost. We also found that AK patients had a higher number of visits and hospitalization rates, likely contributing to their higher total costs. This may be related to the persistent nature of Acanthamoeba [17]. According to studies by Aksozek et al. [18] and Mazur et al. [19], Acanthamoeba cysts are difficult to kill, tenacious in their survival, and not capable of being eliminated by the human immune system completely [18], often causing severe pain in AK patients [17], which may be the fundamental reason for the high number of visits and hospitalization rates. Additionally, AK patients bear the highest registration, ophthalmic exams, and lab test costs, aligning with the characteristics of AK being hard to diagnose and having a high surgical rate [14]. AK diagnosis mainly involves several methods: clinical signs, in vivo confocal microscopy (IVCM), smear, culture, molecular biology, and AS-OCT [20,21]. Currently, IVCM appears to be an appropriate method for diagnosing AK [22]. However, due to their irregular shapes and similarity to corneal stromal cells, diagnosing AK remains challenging [23]. There is still no single effective treatment for AK [24]; broad-spectrum bactericides [25] and personalized treatments based on drug sensitivity tests [26] are the common methods for treating AK. When maximal drug treatment fails to prevent lesion spread to the paracentral corneal stroma, surgical intervention is necessary [27]. Hence, prioritizing early diagnosis, selecting diagnostic approaches with high specificity and sensitivity, and particularly advancing the discovery and development of drugs tailored for AK are crucial steps to alleviate the economic impact on patients with AK. 

The higher medication costs for FK and VK patients may be due to the recurrent nature of VK [24] and the requirement for FK patients to use expensive medications, such as Natamycin [28], costing approximately USD 106.4 per bottle. The recurrent outbreaks of VK might be related to its dormancy within the host’s trigeminal ganglion and reactivation under certain conditions (such as psychological stress, fatigue, immunosuppression, and UV exposure). The reactivated virus then travels back to the eye through the trigeminal ganglion and ophthalmic nerves in a retrograde manner, and the resultant chronic inflammation facilitates corneal damage [29,30]. Consequently, there is an urgent need to develop more cost-effective drugs for FK and VK. According to a survey reported by the American Academy of Ophthalmology [24], vaccination may be a good preventive measure. 

Our study delved into severity’s cost impact, confirming that higher severe MK significantly increases the total costs, echoing previous studies [14], which could be attributed to patients with higher severity levels requiring more frequent visits, thereby driving up costs. There are complex links with medications: FK was initially treated with topical natamycin, yet VK may require oral/intravenous voriconazole [31,32], escalating expenses. Surgery, prompted by worsening ulcer indicators, adds further costs [33], emphasizing early intervention’s importance and necessitating advanced diagnostic capabilities and expertise.

In our study, the presentation time of hospital visits was found as a critical factor influencing the total costs. Patients with a longer time to presentation tended to face higher registration, ophthalmic exams, lab tests, and medication costs [34,35,36], and we emphasize the importance of seeking medical assistance early in managing MK from an economic perspective. An investigation conducted in Nepal [37] demonstrated that indirect hospital routes prolonged presentation times, worsened visual outcomes, enlarged ulcers, and inflated expenses. Thus, swift referral to qualified microbiology-testing hospitals is vital. These insights stress the urgency for public education on timely care-seeking and implementing efficient referral mechanisms to ensure rapid specialist access [38].

Multivariate regression analysis showed that each extra outpatient visit helps to control total costs, whereas each inpatient visit significantly raises total costs, emphasizing the hefty cost of inpatient care and the outpatient setting’s cost-controlling potential. The high cost of inpatient treatment reflects its intensive demand for resources, including advanced medical equipment, intervention of medical professionals, and other medical resources. Hence, optimizing outpatient care for MK can ease patients’ economic load, improve resource use efficiency, and alleviate hospital pressures [39,40]. Early intervention and outpatient strategies, as highlighted, reduce inpatient needs, cutting overall costs. As a result, we see the strengthening of community health, boosting of MK prevention awareness, and adopting of non-/minimally invasive tech as key solutions to reducing the economic burden of MK [41].

This study also has inherent limitations. For example, data from a single center may not fully represent the wider population. As the largest ophthalmic center in North China, most patients have probably visited other hospitals previously, which may affect the number of visits and potentially underestimate cost data. Retrospective studies can only establish correlations, not causality, these limitations are unlikely to significantly impact other variables, or the research. What is more, we did not investigate the impact of microbial resistance on the cost data [42]. Additionally, this article did not account for the indirect costs incurred by MK, which may underestimate the burden MK places on patients.

## 5. Conclusions

In conclusion, this study provides cost data for MK patients in China. The following conclusions can be drawn: AK patients incur the highest total costs, especially in ophthalmic exams and lab tests; FK and VK patients bear higher costs of medication; patients with higher severity levels and longer presentation time spend more; outpatient treatment is significantly cheaper than inpatient treatment.

## Figures and Tables

**Figure 1 pathogens-13-00666-f001:**
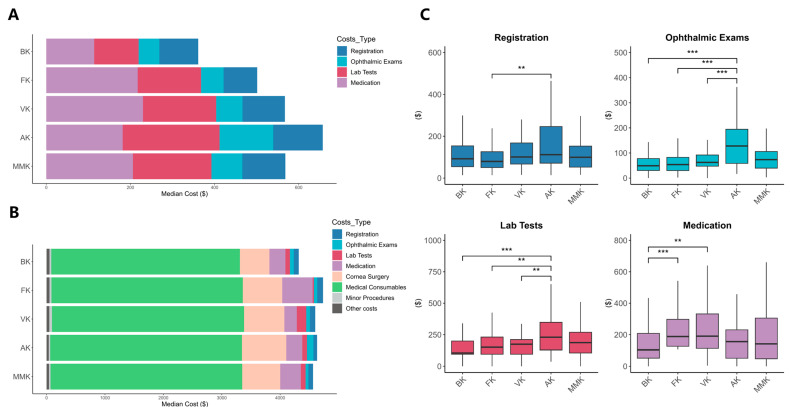
Median costs of patients with different types of keratitis. (**A**) Among outpatient visits; (**B**) among inpatient visits; (**C**) comparative analysis of costs for outpatients with different types of keratitis across different cost types. BK, Bacterial Keratitis; FK, Fungal Keratitis; VK, Viral Keratitis; AK, Acanthamoeba Keratitis; MMK, Mixed Microbial Keratitis. ** represents 0.001 < *p* < 0.01, *** represents *p* < 0.001.

**Figure 2 pathogens-13-00666-f002:**
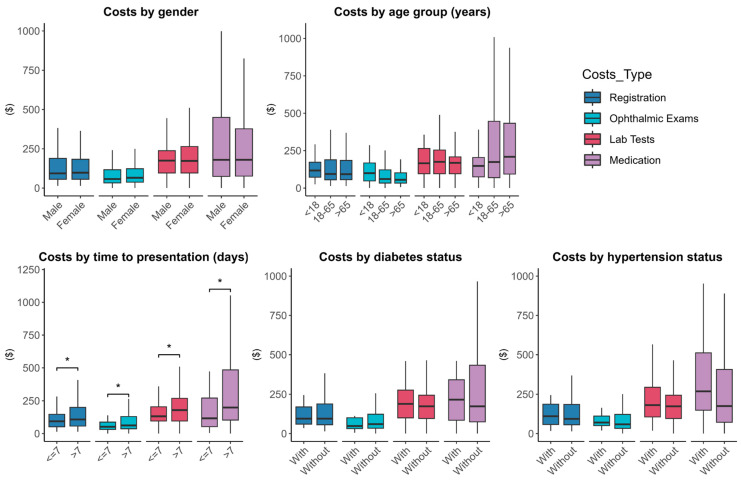
Comparative analysis of different types of costs based on the various influencing factors including gender, age, time to presentation, diabetes, and hypertension. * represents 0.01 < *p* < 0.05.

**Figure 3 pathogens-13-00666-f003:**
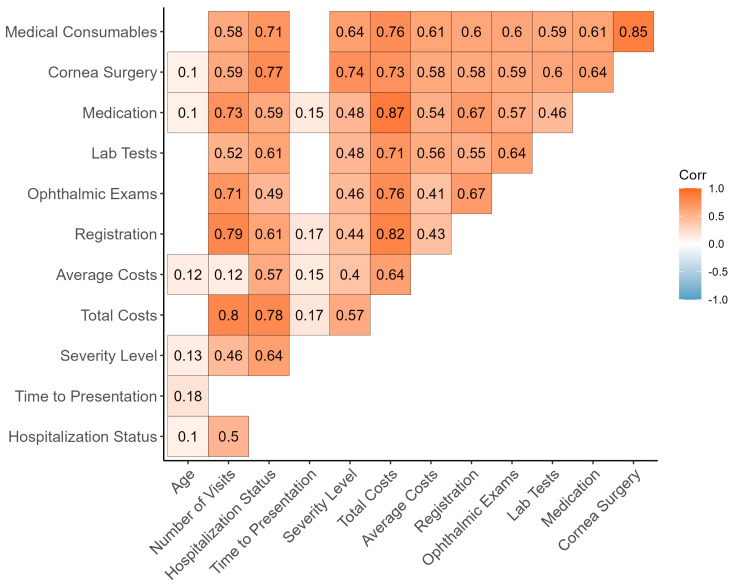
Association measures among demographics, clinical parameters, and cost data of Microbial Keratitis patients (points with *p* > 0.05 have been excluded) using Pearson’s r, Spearman’s rho, and Cramer’s V.

**Table 1 pathogens-13-00666-t001:** The definition of each type of service payment for microbial keratitis treatment.

Categories of Costs	Definition
Registration	Consultation with ophthalmologists
Ophthalmic Exams	Imaging of ocular surface/other ophthalmic tests
Lab Tests	Laboratory microbiological tests
Medication	Medications treatment for outpatient and inpatient individuals
Cornea Surgery	Lamellar keratoplasty, penetrating keratoplasty, etc.
Minor Procedures	Simple treatments such as tear duct flushing, intraocular medication change, venipuncture, etc./nursing care
Medical Consumables	Suturing, surgical instruments/corneal tissue
Other costs	Medical record creation/hospital merchandise

**Table 2 pathogens-13-00666-t002:** Demographics and clinical features of MK patients.

Demographic Factors	Overall (n = 602)	BK (n = 251)	FK (n = 128)	VK (n = 42)	AK (n = 48)	^a^ *p*-Value	MMK (n = 133)	^b^ *p*-Value
**Age (years; median, IQR)**	57 (46–65)	56 (43–65)	58 (51–64)	58 (40–66)	26 (14–54)	<0.001 *	59 (53–68)	<0.001 *
**Gender (n, %male)**	375 (62.3%)	152	88	26	23	0.083	86	0.591
**Time to Presentation (days; median, IQR)**	20 (4–60)	13 (3–60)	18 (8–30)	50 (30–90)	30 (15–90)	<0.001 *	30 (5–60)	0.949
**Number of Visits (median, IQR)**	5 (3–9)	5 (3–8)	5 (3–8)	6 (3–10)	7 (4–12)	0.003 *	6 (4–10)	0.032 *
**Number of hospitalizations**	118 (19.6%)	32	36	5	10	0.002 *	35 (26.3%)	0.037 *
**Laterality**						0.548		0.413
Left eye	294 (48.8%)	124	60	21	25	-	64	-
Right eye	339 (56.3%)	137	71	24	28	-	79	-
Both eyes	31 (5.1%)	10	3	3	5	-	10	-
**Risk Factors**								
Trauma	203 (33.7%)	79	62	3	7	<0.001 *	52	0.167
Contact Lenses	41 (6.8%)	12	1	0	22	<0.001 *	6	0.318
Ocular Disease	198 (32.9%)	103	32	21	6	<0.001 *	36	0.130
Diabetes	35 (5.8%)	18	9	2	2	0.835	4	0.175
Hypertension	30 (5.0%)	14	9	0	1	0.236	6	0.954
**Clinical Symptoms**								
Eye Pain	315 (52.3%)	129	64	18	28	0.529	76	0.245
Blurred Vision	261 (43.4%)	46	13	11	5	0.035 *	15	0.227
Photophobia	70 (11.6%)	29	14	6	7	0.874	14	0.767
**Clinical Signs**								
Corneal Infiltration	334 (55.5%)	131	74	24	35	0.065	70	0.515
Corneal Ulcer	344 (57.1%)	150	70	25	17	0.019 *	82	0.275
Corneal Edema	238 (39.5%)	100	41	19	22	0.229	56	0.558
Neovascularization	149 (24.8%)	68	24	16	10	0.058	31	0.747
Central involvement	301 (50.0%)	109	66	23	29	0.092	74	0.169
Hypopyon	169 (28.1%)	72	38	9	9	0.375	41	0.489

Note: ^a^
*p*-value, calculated through Chi-square test or Kruskal–Wallis test among patients with single microbial keratitis. ^b^
*p*-value, Mann–Whitney U test to compare between patients with single microbial keratitis and those with mixed microbial keratitis. BK, Bacterial Keratitis; FK, Fungal Keratitis; VK, Viral Keratitis; AK, Acanthamoeba Keratitis; MMK, Mixed Microbial Keratitis. * = significant *p*-value < 0.05.

**Table 3 pathogens-13-00666-t003:** Total and average costs of outpatients with microbial keratitis at varying severity levels.

Type of Keratitis	Overall	Level 1–3	Level 4	*p*-Value
**Bacterial Keratitis**				
Number of Cases (%)	251 (41.7%)	180	71	-
Total Costs	419.0 (281.4–750.2)	355.8 (246.0–520.9)	878.2 (523.1–1586.0)	<0.001 *
Average Costs	95.1 (70.4–133.4)	89.6 (65.7–123.4)	102.6 (86.4–156.6)	0.002 *
**Fungal Keratitis**				
Number of Cases (%)	128 (21.3%)	73	55	-
Total Costs	571.5 (376.5–1031.9)	418.3 (315.9–572.8)	930.6 (632.6–1598.3)	<0.001 *
Average Costs	118.5 (93.2–161.5)	114.2 (91.9–156.3)	124.9 (93.9–169.3)	0.313
**Viral Keratitis**				
Number of Cases (%)	42 (7.0%)	29	13	-
Total Costs	577.1 (377.1–927.4)	547.4 (299.6–676.2)	728.1 (422.9–1619.4)	0.097
Average Costs	97.8 (75.83–153.4)	95.3 (63.3–143.6)	145.6 (81.1–187.4)	0.182
**Acanthamoeba Keratitis**				
Number of Cases (%)	48 (8.0%)	29	19	-
Total Costs	703.7 (399.2–1380.3)	524.3 (371.1–706.4)	1421.3 (990.3–2329.5)	<0.001 *
Average Costs	97.4 (73.5–126.4)	91.7 (70.8–104.1)	121.6 (82.9–173.9)	0.084
**Mixed Microbial Keratitis**				
Number of Cases (%)	133 (22.1%)	67	66	-
Total Costs	642.1 (352.9–1137.3)	472.6 (283.1–637.4)	867.6 (645.9–1554.4)	<0.001 *
Average Costs	111.1 (78.8–147.8)	90.7 (73.2–145.9)	127.0 (93.4–149.1)	0.005 *

Note: *p*-value, calculated through Mann–Whitney U test to compare between patients whose severity levels are 1–3 and 4. Costs data are presented in USD as median (IQR). * = significant *p*-value < 0.05.

**Table 4 pathogens-13-00666-t004:** Direct costs of microbial keratitis influenced by various factors.

Influencing Factors	Number (%)	Total Costs (USD; Median, IQR)	*p*-Value
**Gender**			
Male	375 (62.3)	553.6 (328.8–1648.6)	0.679
Female	227 (37.7)	548.0 (340.0–992.4)	
**Age (years)**			
<18	43 (7.1)	606.0 (340.5–752.6)	0.758
18–65	415 (68.9)	514.7 (332.2–1444.7)	
>65	144 (23.9)	611.4 (339.0–1189.9)	
**Time to Presentation (days)**			
≤7	99 (19.4)	385.2 (298.3–686.1)	<0.001 *
>7	410 (80.6)	600.3 (352.7–1660.7)	
**Diabetes**			
With	35 (5.8)	664.4 (409.5–797.1)	0.612
Without	567 (94.2)	540.0 (332.2–1287.1)	
**Hypertension**			
With	30 (5.0)	749.7 (451.1–1717.2)	0.118
Without	572 (95.0)	540.7 (331.1–1210.5)	

Note: *p*-value, calculated through Mann–Whitney U test (Kruskal–Wallis test for age group) to compare under different influencing factors. * = significant *p*-value < 0.05.

**Table 5 pathogens-13-00666-t005:** Multivariate linear regression analysis of factors influencing total costs.

Parameter	Estimate ($)	Standardized Estimate	95%CI of SE	*p*-Value
Acanthamoeba keratitis	499.310	0.065	(0.022, 0.385)	0.028 *
Number of outpatient visits	100.290	0.245	(0.173, 0.298)	<0.001 *
Number of inpatient visits	2759.546	0.595	(0.507, 0.649)	<0.001 *
Severity level = 4	552.675	0.107	(0.077, 0.373)	0.003 *
Time to presentation > 7 days	311.634	0.052	(−0.014, 0.268)	0.078

Note: adjusted R^2^ = 0.672, *p* < 0.001. All values in the “Estimate” column are presented in US dollars. * = significant *p*-value < 0.05.

## Data Availability

The information utilized in our research can be obtained upon request directed to the corresponding authors. Due to the necessity of preserving patient privacy, the dataset is not openly accessible to the public domain.

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
