# Peer review of "Direct Cost Analysis of Microbial Keratitis in North China: A Hospital-Based Retrospective Study"

_pathogens, 2024, doi:10.3390/pathogens13080666_

Round 1

Reviewer 1 Report

Comments and Suggestions for Authors

Shi et al. conduct an economic analysis of infectious keratitis in northern China. This topic holds general interest due to the importance of quantifying the costs of all infectious diseases, even those that appear minor, as it informs the selection of optimal diagnostic algorithms to maximize cost-effectiveness from both health and social perspectives. However, I believe the article requires improvements in certain methodological aspects to achieve the quality necessary for broader relevance.

Major Comments:

The reader needs a clear understanding of how the costs were calculated. The authors should provide a more detailed explanation of several aspects:

The characteristics of the infections and their treatments: What proportion of infections were caused by NFGNB, Enterobacteriaceae, or gram-positive cocci? How many cases were due to adenoviruses, herpes simplex, etc.? What were the main characteristics of the antimicrobial or medical treatments? It is common to use specially formulated eye drops from intravenous antibiotics, which involve additional costs due to hospital pharmacy interventions. What was the antimicrobial therapy used for treating Acanthamoeba? What microorganisms constitute mixed microbial keratitis? What PCR methods were used for diagnosis?

The clinical outcomes of these interventions: Were there any treatment failures necessitating corneal transplantation or resulting in the loss of the eye? In summary, all clinical and microbiologic findings, as well as treatment characteristics, need to be thoroughly explained.

The source of the cost data needs to be better detailed. Simply stating "the cost of diagnosis and treatment was $577" is insufficient. The methodology behind this calculation must be clarified.

Minor Comment:

The tables should be self-explanatory. All abbreviations used in the tables should be explained within the table itself, ideally as footnotes

Reviewer 2 Report

Comments and Suggestions for Authors

Congratulations on your excellent analysis of the treatment costs for microbial keratitis and its clinical implications. Your thorough examination provides valuable insights into the economic burden of this condition and highlights the importance of cost-effective treatment strategies. The depth of your research not only enhances our understanding of the financial aspects but also emphasizes the need for optimized clinical practices to improve patient outcomes. Your work is a significant contribution to the field and will undoubtedly aid in better healthcare decision-making. Well done!

just some minor comments:

- how many persons extracted and analyzed the data? one or more? and if any differences was obtained, how do you managed it?

- line 264-265: the value of AS-OCT is also being studied in different infections, please add it

- you should add an analysis about microbial resistance of the different infections, as more the resistance more the median time to recover and thus the cost. If you don't have this data, add in the limitation and add in the discussion the reference to it [D'Oria F, Buonamassa R, Rizzo T, Boscia F, Alessio G, Guerriero S. Bacterial isolates and antimicrobial susceptibility pattern of ocular infection at a tertiary referral hospital in the South of Italy. Eur J Ophthalmol. 2023;33(1):370-376. doi:10.1177/11206721221106139]

Comments on the Quality of English Language

minor editing

Round 2

Reviewer 1 Report

Comments and Suggestions for Authors

The authors have correctly answered most of the questions; however, some aspects remain to be clarified..

Major Comments

The origin of the costs remains unclear. In the Materials and Methods section, the authors list the different components of the total cost, specifically: Registration, Ophthalmic Exams, Lab Tests, Medication, Cornea Surgery, Minor Procedures, and Medical Consumables. However, the results for each of these expenses are not shown, either in the text or, preferably, in a table.

  • For Ophthalmic Exams, the average costs for AK, VK, and BK are X1, X2, and X3, respectively. For Lab Tests, the average costs for AK, VK, and BK are... In this way, the reader can trace the origin of the total average costs for each pathology.

Minor Comments

  • Typography of Microorganisms: Microorganisms should be written in italics, and if only the genus is mentioned, use "spp."
    • FusarioumFusarium spp
    • Acanthamoeba spp
  • Figure 4 (Corr): In Figure 4, the value labeled as "Corr" should be clarified. I understand this represents Spearman's rho value. Use the term "rho" or "Spearman's rho" instead of "Corr."
  • Table 5: The "Estimate" column lacks units. I assume these are in dollars, but tables and figures need to be self-explanatory. Use the symbol ($).
  • In the Discussion section, the sentence "Multivariate regression analysis showed that each extra outpatient visit notably raises total costs, whereas each inpatient visit has a more substantial effect, emphasizing both the hefty cost of inpatient care and the outpatient setting's cost-controlling potential." seems to be written in reverse. Should it be corrected?
